# Impact of SchisandraChinensis Bee Pollen on Nonalcoholic Fatty Liver Disease and Gut Microbiota in HighFat Diet Induced Obese Mice

**DOI:** 10.3390/nu11020346

**Published:** 2019-02-06

**Authors:** Ni Cheng, Sinan Chen, Xinyan Liu, Haoan Zhao, Wei Cao

**Affiliations:** 1College of Food Science and Technology, Northwest University, 229 North TaiBai Road, Xi’an 710069, China; chensinan111@163.com (S.C.); liuxinyan946926@163.com (X.L.); 2School of Chemical Engineering, Northwest University, 229 North TaiBai Road, Xi’an 710069, China; haoan_zhao@126.com; 3Bee Product Research Center of Shaanxi Province, Xi’an 710065, China

**Keywords:** *Schisandrachinensis*bee pollen, gut microbiota, nonalcoholic fatty liver disease, obesity, phenolic compounds

## Abstract

*Schisandrachinensis*bee pollen has been used as a health food in China for centuries; however, its bioactive constituents and functions are not very clear. In this study, we investigated the phenolic compounds of *Schisandrachinensis*bee pollen extract (SCPE) by UHPLC-Q-Orbitrap-HRMS/HPLC-DAD-ECD and its prevention from nonalcoholic fatty liver disease (NAFLD) and modulation of gut microbiota in high fat diet induced obese C57BL/6 mice. The results showed that 12 phenolic compounds were identified in SCPE, and naringenin, rutin and chrysin were the main constituents. The content of naringenin reached 1.89 mg/g, and total phenolic content (TPC) of SCPE were 101.83 mg GA/g. After obese mice were administrated with SCPE at 7.86 and 15.72 g/kg BW for 8 weeks, body weight gains were reduced by 18.23% and 19.37%. SCPE could decrease fasting blood glucose, cut down the lipid accumulation in serum and liver, lessen oxidative injury and inflammation in obesity mice. Moreover, SCPE could effectively inhibit the formation of NAFLD by inhibition of LXR-α, SREBP-1c and FAS genes expression, and modulate the structural alteration of gut microbiota in obesity mice. These findings suggested that SCPE could attenuate the features of the metabolism syndrome in obesity mice, which can be used to prevent obesity and NAFLD of human beings.

## 1. Introduction 

Obesity is a growing serious public health problem in China and worldwide associated with chronic metabolic diseases such as type 2 diabetes, coronary artery disease, hypertension, hyperlipoidemia, nonalcoholic fatty liver disease (NAFLD), and even cancers in China and all over the world [1,2,3]. Recent reviews have demonstrated that a growing number of children and adolescents suffer from overweight and obesity, seriously affectingtheir physical and intellectual development [4,5,6].NAFLD is a common disease induced by obesity in children and adolescents [7], accompanied by excessive fat deposition in hepatocytes inthe absence of significant alcohol consumption [8].Dietary fat, especially saturated fatty acids, should be one of the important factors in the development of NAFLD [9]. Usually, NAFLD accompanied with inflammation, fibrosis and malignant transformation was difficult to diagnose in the early stage [10]. Therefore, prevention and dietary intervention are becoming increasingly important in people’s daily lives. 

Phenolic compounds are a large group of plant secondary metabolites, which possess the capabilities of antioxidant, anti-inflammation, antitumor, antihypertensive, and prevention of the diseases induced by oxidative stress [11]. Recent studies have reported that phenolic compounds can reach the colon, be utilized by gut microbiota, and take part in nutrient acquisition and lipid metabolism,playing a critical role in metabolic diseases [12,13]. Tea polyphenols could modulate gut dysbiosis induced by high-fat diet and causeweight loss in mice [14,15,16]. Polyphenol-rich extract of pomegranate peel consumption regulated gut microbiota and then alleviated tissue inflammation and hypercholesterolaemia in obese mice [17]. Grape polyphenolic consumption altered gut microbiota and reduced adiposity [18]. Lychee pulp polyphenols reduced the body weight and ameliorate hepatic lipid accumulation in obese mice induced by high-fat diet [19]. However, there has beenno research on the anti-obesity effect and prevention from NAFLD of bee pollen polyphenols conducted to date. 

*Schisandrachinensis*bee pollen collected by honey bees from the flowers of *S. chinensis* is a widely cultivated plant in China and has been used as medicine to treat liver disease for several centuries. In our previous study, healthy protection of *S. chinensis*bee pollen has been studied, the result elucidated that *S. chinensis* pollen possessed strong antioxidant activities and prevented liver damage induced by CCl_4_. However, there has beenno research on the relationship of *S. chinensis* pollen and obesity, NAFLD. In this study, we investigated the phenolic compounds of *S. chinensis* pollen extracted (SCPE) by UHPLC-Q-Orbitrap-HRMS/HPLC-DAD-ECD, and its antioxidant activities in vitro. More importantly, administration with SCPE preventingNAFLD and gut microbiota disorder induced by high-fat diet in mice were studied in this report, providing a theoretical reference for the biological function of *S. chinensis*bee pollen.

## 2. Materials and Methods

### 2.1. Preparation and Identification of SCPE 

Bee pollen of *S. chinensis* was purchased from Hubei province. Its floral origin was identified by comparison with pollen morphology of S. *chinensis* reported by Ai et al. [20] and Sun [21]. The sample was extracted twice with 10 times 75% (*v*/*v*) ethanol under heat refluxfor 2 h at 75 °C. The supernatant was collected by centrifugation at 4800 rpm for 10 min and concentrated under vacuum to give the *S. chinensis* pollen extract (SCPE).

UHPLC-Q-Orbitrap-HRMS analysis of SCPE wasperformed using aUHPLC system (1290, Agilent Technologies) with UPLC HSS T3 column (2.1 mm × 100 mm, 1.8 μm) coupled to Q Exactive (Orbitrap MS, Thermo). The mobile phase A was 5 mmol/L ammonium acetate in water for negative, and the mobile phase B was acetonitrile. The elution gradient was set as follows: 0 min, 1% B; 1 min, 1% B; 8 min, 99% B; 10 min, 99% B; 10.1 min, 1% B; 12 min, 1% B. The flow rate was 0.5 mL/min. The injection volume was 1 μL. MS data were executed in negative ion mode because of one or more carboxylic acid or hydroxyl groups containing in polyphenols. HPLC-DAD-ECD (U3000, Thermo Fisher) was used in the quantitative analysis of the phenolic compounds in SCPE. All the standards of phenolic compounds in this study were chromatographic grade and purchased from Sigma-Aldrich (USA). Due to the complexity of polyphenols in SCPE, total phenolic content (TPC) and total flavonoid content (TFC) were also determined by the methodsused in our previous study [22]. The result of TPC was calculated with the gallic acid equivalents (mg GA/g SCPE) and TFC was calculated with rutin equivalents (mg rutin/g SCPE) respectively.

### 2.2. Antioxidant Activities of SCPE 

DPPH radical scavenging activity was measured according to the method proposed by Brand-Williams (1995) [23]. Briefly, different volumes of SCPE solution at 1.0 mg/mL were mixed with 5.0 mL of a methanolic solution of DPPH radicals at 0.1 mM in brown test tubs. The total volume was adjusted to 10 mL with methanol, and then shaken evenly. It was stand at room temperature in the dark for 30 min. Absorbance was measured at 517 nm. The result was expressed as IC_50_ value (the concentration of SCPE required to reach the inhibition of DPPH radicals to 50%). 

Ferric reducing antioxidant power (FRAP) of SCPE was determined according to a modified method [24]. SCPE solution was mix with 4.0 mL of FRAP reagent (2.5 mL of 20 mM FeCl_3_·6H_2_O and 25 mL of 0.3 M acetate buffer, pH 3.6). The absorbance was measured at 593 nm and the result was expressed as Trolox equivalent (mg Trolox/g SCPE).

Ferrous ion-chelating activity of SCPE was carried out by the method proposed by Singh and Rajini [25]. The absorbance of ferrozine-Fe^2+^ complex was determined at 562 nm. Ferrous ion-chelating activity was expressed as Na_2_EDTA equivalents (mg Na_2_EDTA/g SCPE).

### 2.3. Animal Experiments 

Forty-eight male C57BL/6 mice (16-18 g) were obtained from laboratory animal research center of Fourth Military Medical University (Xi’an, Shaanxi, China), and the license number was SCXK (Army) 2012-007. All mice were raised under standard laboratory conditions (temperature 25 ± 2 °C, relative humidity 50 ± 10%, under 12 h light-dark cycles). After acclimation for 1 week, the mice were divided into four groups, randomly. One group was provided with a low-fat diet (LFD, 8% fat, 22% proteins, 70% carbohydrates), which supplied 3.4 kcal/g energy. The other three groups were provided with a high-fat diet (HFD, 45% fat, 18% proteins, 37% carbohydrates), which supplied 5.2 kcal/g energy. LFD and HFD were provided until the end of the experiment. All the feed was purchased from Shanghai Fanbo Biotechnology Co., Ltd. (Shanghai, China). Eight weeks later, two of the HFD groups were respectively given 7.86 and 15.72 g SCPE/kg BW byintragastricadministration for the following 8 weeks. The four groups in this study were expressed as LFD, HFD, HFD + LE (7.86 g/kg BW) and HFD + HE (15.72 g/kg BW) groups, respectively. 

Fasting body weight was obtained every two weeks. At the end of 16th week, all the mice were fasted for 10 h and then blood samples were obtained by cardiac puncture. A drop of blood was used for blood-glucose testing by a blood glucose meter (Yicheng Bioelectronics Technology Co., Ltd., Beijing, China). After killing the mice by cervical dislocation, liver, spleen, colon, and epididymal adipose were separated and the liver and spleen were weighed out immediately. 

### 2.4. Serum and Hepatic Biochemical Analysis 

The serum samples were analyzed for total cholesterol (TC), triglyceride (TG), LDL-cholesterol (LDL-C), alanine aminotransferase (ALT), aspartate aminotransferase (AST) using commercially available diagnostic kits produced by Nanjing Jiancheng Bioengineering Institute (Nanjing, China). Serum interleukin-6 (IL-6), tumor necrosis factor α (TNF-α) were characterized by immunoassay using commercial ELISA kits produced by Shanghai Fusheng Industrial Co., Ltd. (Shanghai, Chian). All the test procedures were carried outaccording to the manufacturer’s instructions.

A part of the liver was homogenized in 9 volumes of ice-cold physiological saline, centrifuged at 2500 rpm for 15 min at 4 °C. The supernatant was used for following analysis. Diagnostic kits from Nanjing JianchengBioengineering Institute (Nanjing, China) were used in the determination of malondialdehyde (MDA), superoxide dismutase (SOD) and glutathione peroxidase (GSH-Px) activity, and the ELISA kit for adiponectin purchased from Shanghai Fusheng Industrial Co., Ltd. (Shanghai, China) was used for quantifying the adiponectin in liver. The results were normalized with protein. In addition, anhydrous alcohol served as the homogenate medium and the supernatant was used for determination the content of TG and TC. The results expressed as the content per liter supernatant (mmol/L). 

### 2.5. Quantification of Gene Expression Analysis 

Total RNA from liver was extracted using a Takara minibest universal RNA extraction kit (Takara, Dalian, China) according to the manufacturer’s instructions. RNA was reverse-transcribed using Primescript^TM^rt master mix kit (Takara, Dalian, China). The cDNA samples were amplified in duplicate. Quantitative real-time PCR (Q-PCR) was performed on the Real Time PCR amplification instrument (Gentier 96E, Xi’an Tianlong Science and Technology Co., China) with TB Green^TM^ Premix Ex Taq^TM^ II. The Q-PCR conditions were as follows: 95 °C, 30 s, followed by 40 cycles at 95 °C, 5 s, 60 °C, 30 s. The primer sequences of interleukin -1β (IL-1β), TNF-α, nuclear factor κB (NF-κB), inducible nitric oxide synthase (iNOS) and β-actin were designed according to the method proposed by Wu et al. (2015) [26]. The primer sequences of fatty liver related genes such as liver X receptors (LXR)-α, sterol regulatory element binding protein (SREBP)-1c and fatty acid synthase (FAS) were designed according to the study of Sim et al. (2014) [27]. Purity of PCR products was assessed by melt curve analysis. Gene expression was examined and normalized with β-actin, which served as an internal control in PCR, and the relative fold induction was calculated using the formula 2^-△△Ct^ [28]. 

### 2.6. Histopathological Examinations of Liver and Epididymal Adipose

For the histological evaluation, a portion of liver and epididymal adipose were fixed in 10% neutral formalin and embedded in paraffin wax. Sections of 5 μm thickness were cut, deparaffinized, dehydrated, and then stained with hematoxylin and eosin (H&E). 

### 2.7. Illumina Sequencing and Statistical Analysis of 16S rRNA Gene V3-V4 Region of Gut Microbiota 

Total genomic DNA was extracted from colon contents using the Fast SPIN extraction kits (MP Biomedicals, Santa Ana, CA, USA), according to the manufacturer’s instructions. The quantity and quality of extracted DNA were measured using a NanoDrop ND-1000 spectrophotometer (Thermo Fisher Scientific, Waltham, MA, USA) and agarose gel electrophoresis, respectively. PCR amplification of the bacterial 16S rRNA genes V3-V4 hypervariable regions was performed using the forward primer 338F (5′-ACTCCTACGGGAGGCAGCA-3′) and the reverse primer 806R (5′-GGACTACHVGGGTWTCTAAT-3′) according to the method proposed by Liu et al. (2018) [29]. PCR amplicons were purified with AgencourtAMPure Beads (Beckman Coulter, Indianapolis, IN) and quantified using the PicoGreendsDNA Assay Kit (Invitrogen, Carlsbad, CA, USA). After the individual quantification step, amplicons were pooled in equal amounts, and pair-end 2×300 bp sequencing was performed using the IllluminaMiSeq platform with MiSeq Reagent Kit v3 at Shanghai PersonalBiotechnology Co., Ltd (Shanghai, China). 

The quantitative insights into microbial ecology (QIIME, v1.8.0) pipeline was employed to process the sequencing data according to the method proposed by Caporaso et al. (2010) [30]. The operational taxonomy units (OTUs) of representative sequences at 97% similarity and their relative abundance were used to calculate rarefaction analysis and Shannon diversity index by UCLUST. Nonmetric multidimensional scaling (NMDS) were used to examine the abundance and diversity of the OTUs.

### 2.8. Statistical Analysis 

All the tests were performed in triplicate. Statistically significant differences were evaluated by Duncan’s multiple range test after SAS one-way ANOVA, version 8.1 (SAS Institute, Cary, NC, USA). Differences at *p* < 0.05 were considered to be significant.

## 3. Results

### 3.1. The Phenolic Compound and Antioxidant Activity of SCPE

Twelve phenolic compounds were identified in SCPE by UHPLC-Q-Orbitrap-HRMS based onaccurate comparison of experimental ions, calculated ions, and fragment ions. Seven phenolic compounds were quantified by HPLC-DAD-ECD, and the results areshown in Table 1. Naringenin was the most abundant compound in the quantified phenolics, reaching 1.89 mg/g. Rutin and chrysin were more than 0.5 mg/g in SCPE. The structural formulas of all identified phenolic compounds areshown in the Appendix A.

The TPC, TFC and antioxidant activity in vitro of SCPE aredisplayed in Table 2. The TPC and TFC were 101.83 mg GA/g and 73.22 mg rutin/g, respectively. DPPH scavenging activity (IC_50_) was 0.74 mg/mL. The FRAP value was 297.80 mg Trolox/g. Ferrous ion-chelating activity was 44.82 mg Na_2_EDTA/g.

### 3.2. SCPE Attenuated Obesity Induced by HFD in Mice 

Obesity and type 2 diabetes are the two major risk factors for NAFLD;therefore, body weight, liver weight and fasting blood glucose were determined in this study and the results were shown in Figure 1. The mice were weighedevery two weeks during the experiment. At the end of 8th week, the weight of the mice with HFD reached 33.94 g, which was an increase of31.50% in comparison to theLFD group. After administration with SCPE at adose of 7.86 and 15.72 g/kg BW, the body weight declined by 18.23% and 19.37% over the HFD group, respectively. The liver weight of the HFD group was significantly higher than LFD group (*p* < 0.05). Excitingly, SCPE consumption completely inhibited the increase of liver weight induced by HFD. Similarly, the increase of fasting blood glucose induced by HFD was ameliorated by SCPE administration. There was no significant difference in spleen weights among all mice.

### 3.3. Effects of SCPE on TC, TG and LDL-C Levels in Serum and Liver 

As shown in Figure 2A,B, a 201% increase of TC level and a 45.1% increase of TG level in serum was observed in the HFD group. Treatment with SCPE didnot decrease the content of TC in serum. However, SCPE administration in different dosagescould significantly decrease the content of TG in serum (*p* < 0.05). Although HFD did not increase serum LDL-C level, SCPE consumption decreased serum LDL-C level compared to the HFD group (*p* < 0.05). 

The contents of TC and TG in the liver were significantly increased by HFD feeding (*p* < 0.05) (Figure 2C). A 74.1% increase of TC content and a 107% increase of TG content in liver were observed in the HFD group. Surprisingly, SCPE administration significantly attenuated the increase of TC and TG contents in liver with a dose-effect relationship (*p* < 0.05). Treatment with 15.72 g/kg BW of SCPE could completely inhibit the increase of TC content in liver induced by HFD. There was no difference between the LFD and HFD + HE groups (*p > 0.05*). SCPE administration at doses of 7.86 and 15.72 g/kg BW decreased TG contents in liver by 20.1% and 35.1% compared with the HFD group (*p* < 0.05). 

### 3.4. Effects of SCPE on Liver Damage, Oxidative Stress, Inflammation and NAFLD Formation 

Serum aminotransferase is usually used to assess hepatitis [31]. In this study, as shown in Figure 3A, ALT activity was significantly increased by HFD (*p* < 0.05). ALT activity in the HFD group was 28.15 U/L, 1.66 times that of theLFD group. SCPE at 7.86 and 15.72 g/kg BW could mitigatethe increase of ALT activity induced by HFD (*p* < 0.05). More importantly, administration with 15.72 g/kg BW of SCPE could completely counteract the increase of ALT activity. There was no difference between the LFD group and the HFD+HE group (*p > 0.05*). Serum AST activities werenodifferent between all mice (*p > 0.05*). 

A previous study reported that oxidative stress occurs in obesity and NAFLD induced by HFD [32]. Therefore, hepatic oxidative stress was determined, and the results are shown in Figure 3B–D. A 21.5% decrease of SOD activity and a 26.5% decrease of GSH-Px activity were observed in the HFD group compared with the LFD group (*p* < 0.05). However, administration of 7.86 and 15.72 g/kg BW of SCPE for 8 weeks completely inhibited the HFD-induced decrease inGSH-Px activity. Treatment with 15.72 g/kg BW SCPE significantly increased the SOD activity compared with the HFD group (*p* < 0.05). MDA, a product of lipid peroxidation, was measured in this study. An 87.5% increase of MDA content in liver of HFD mice (model group) was observed compared with LFD mice (control) (*p* < 0.05). Importantly, administration with SCPE completely inhibited MDA productioncompared with the HFD group (*p* < 0.05). A diponectin associated with the metabolism of hepatic lipid [32] was determined in this study. As shown in Figure 3E, hepatic adiponectin content was markedly decreased by HFD, showing about a 45.96% decline compared with the LFD group (*p* < 0.05). Compared with the HFD group, SCPE consumption at doses of 7.86 and 15.72 g/kg BW helped to increase the hepatic adiponectin secretion by 46.13% and 75.86%, respectively (*p* < 0.05).

Serum inflammatory cytokine IL-6 and TNF-α were determined by ELISA kits, and the results areshown in Figure 4A,B. Compared to theLFD group, HFD induced a 16.3% increase of IL-6 level and a 30.4% increase of TNF-α level (*p* < 0.05). Meanwhile, 7.86 and 15.72 g/kg BW of SCPE consumption induced a 25.9% and 33.0% decrease of IL-6 level, 21.2% and 15.2% decrease of TNF-α level compared with the HFD group, respectively (*p* < 0.05).

Q-PCR analysis was performed to evaluate the expression of IL-1β, TNF-α, NF-κB and iNOS in the liver (Figure 4C). Mice fed with HFD showed significant increases in TNF-α, IL-1β, NF-κB and iNOS levels compared with LFD mice. SCPE consumption at 7.86 and 15.72 g/kg BW significantly inhibited the expression levels of TNF-α, NF-κB and iNOS(*p* < 0.05). The increase of the expression level of IL-1β wasn’t inhibited by 7.86 g/kg BW of SCPE consumption. However, 15.72 g/kg BW of SCPE consumption reduced the IL-1β expression by 35.67% compared to the HFD group (*p* < 0.05). 

LXR-α, SREBP-1c and FAS play important roles in lipogenesis and the formation of NAFLD [33];therefore, these genes were monitored in this study. In the HFD group, LXR-α, SREBP-1c and FAS gene expressions were all at significantly higher levelscompared to thosein the LFD group (*p* < 0.05, Figure 4D). Surprisingly, administration ofSCPE at 7.86 and 15.72 g/kg BW significantly inhibited the expressionof these genes (*p* < 0.05), whereby 15.72 g/kg BW of SCPE consumption inhibited LXR-α, SREBP-1c and FAS gene expression by 31.65%, 63.59%, 77.77%, respectively. 

Morphological analyses of epididymal adipose and liver were conducted, and the results areshown in Figure 4E,F. The adipocytes of the HFD group were bigger than the LFD group. In addition, administration of SCPE inhibited the increase of adipocytes induced by HFD. As shown in Figure 4F, LFD-fed mice showed normal hepatic tissue with regular morphology of hepatic cells. After a 16-week HFD feeding, serious fatty liver disease occurred, manifesting as macrovesicularsteatosis, cell ballooning degeneration and even hepatic parenchyma reticularis. Administration of SCPE effectively reversed the pathological changes induced by HFD in dose-effect relation. In HFD + LE group, few large lipid vacuoles were observed, and small vacuoles still existed in several hepatic cells. In the pathological section from the HFD+HE group mice, a few small vacuoles were observed. This was nearly same as for theLFD group. 

### 3.5. SCPE Consumption Modulates Gut Microbiota in Obese Mice 

Pyrosequencing of variable region V3-V4 of bacterial 16S rRNA genes was carried out in this study to evaluate the impact of SCPE on mice gut microbiota. We produced a total of 585,256 sequences, with an average of 47,249 sequences in LFD group, 44,504 sequences in the HFD group, and 45,772 (HFD+LE) and 33,252 (HFD+HE) sequences in the SCPE groups, respectively. Theses sequences had a length of 420-450 base pairs (Figure 5I). As shown in the Venn diagram (Figure 5II), the qualified sequences (>0.001%) were clustered into 6143 bacterial OTUs and 841 same OTUs were identified in LFD and HFD groups, 1062 same OTUs were found in LFD and HFD + LE groups, and 1028 same OTUs were found in LFD and HFD+HE groups. Administration with SCPE increased the same OTUS in LFD mice and HFD mice. The rarefactioncurve (Appendix A) showed that the current sequencing depth in each sample was enough for analysis of the gut microbial diversity. The chao1 and Shannon curves showed that the abundance and variety of gut microbiota in all studied mice could be observed, and the following determination and analysis results were reliable. 

To explore the similarity of gut microbial communities in different groups, Nonmetric Multidimensional Scaling analysis based onUniFrac distance was conducted, and the resultsareshown in Figure 6I. The distance matrix of different samples described the similarity. Five samples in the LFD group clustered, and samples in the HFD group were far away from the LFD group. This indicated that high-fat diet induced significant gut microbial diversification. SCPE consumption at 15.72 g/kg BW (HFD + HE group) could shift the gut microbiota composition disrupted by HFD. The effect of SCPE on the key community of gut microbiotaisshown in Figure 6II. *Bacteroidia, Bacteroidetes, Bacteroidates* and *S24-7*, etc., were the main communities in the gut of LFD mice. HFD induced a significant effect on the species including *Proteobacteria*, *Deltaproteobacteria*, *Desulfovibrionaceae* and *Desulfovibrionales*, etc. Different doses of SCPE consumption demonstrated a selective enrichment of *Coriobacteriales*, *Coriobacteriia*, *Coriobacteriaceae*, and *Bifidobacterium*, *Bifidobacteriales*, *Bifidobacteriaceae*, respectively. The effect of SCPE treatment on the relative abundance of bacterial at family level isshown in Figure 6III. *Lactobacillaceae, S24-7* and*Verrcomicrobiaceae* were the main gut microbiota at the family level, whereby*Lactobacillaceae* were remarkably reduced by HFD, and SCPE attenuated the reductionwith a significant dose-effect relationship. To better understand the gut microbial differences among four groups, violin plot and box plot were used in this study (Figure 6IV). The abundance of *Actinobacteria*, *Proteobacteria* and *Pseudomonas* were low in LFD group. However, HFD induced significant increase of those. SCPE consumption could significantly attenuate the increases compared to HFD group. The abundance of *Lactobacillu*, which is a kind of probiotic, was high in the LFD group. Nevertheless, it was greatly reduced by HFD. Treatment with SCPE could increase the *Lactobacillu* in the colon with a dose-effect relationship. Those gut microbial communitiescan protect health mainly by regulating the metabolism of carbohydrate, amino acid and energy (Appendix A). 

## 4. Discussion

Obesity has been a worldwide topic because it is associated with many diseases, including hyperglycemia, cardiovascular disease and metabolic syndrome, which seriously affect people’s health [4,32,34]. Obesity induced by high-fat diet (HFD) is accompanied by not only the body weight increase, adipose deposition in liver and other organs [25,35], but also oxidative stress and inflammation increase, such as IL-1β, TNF-α and iNOS [32]. As expected, the present investigation showed that HFD induced the following: an obvious gain inbody weight and liver weight, an increase of fasting blood glucose, elevations of lipid levels in serum and liver, increases of serum ALT activity and hepatic MDA content, a significant upregulation of the expression of IL-1β, TNF-α, NF-kB and iNOS, decreases of SOD and GSH-Px activities, inhibition of adiponectin secretion, and upregulation of NAFLD-related gene expression, such as LXR-α, SREBP-1c and FAS. Interestingly, after treatment of obese mice with SCPE for 8 weeks, all of the above variations were alleviated. The results indicate that SCPE can inhibit the obesity induced by HFD and prevent NAFLD. 

Usually, obesity accompanied by oxidative stress due to reactive oxygen species (ROS) can be produced duringmitochondrial and peroxisomal fatty acid oxidation. Another way to produce ROS is over-consumption of oxygen, which generates ROS in the mitochondrial respiratory chain coupled with oxidative phosphorylation in the mitochondria [32]. When oxidative stress persists for a long time in an obese body, antioxidants can be depleted, and antioxidant enzymes activities can be decreased [36]. In the present study, the activities of SOD and GSH-Pxwere decreased and MDA content was increased in HFD mice. Administration ofSCPE to obesemice for 8 weeks significantly increased the activities of SOD and GSH-Px and decreased the MDA content compared to the HFD group. Additionally, there is a huge amount of fat deposited all over the body in animals with obesity. When deposition of liver fat exceeds5% of hepatocytes inthe absence of significant alcohol intake, NAFLD can be formed [37]. The nuclear receptor LXR-α has been implicated in the regulation of lipogenesis and cholesterol homeostasis [38]. Activation of LXR-α can result in the development of hepatic steatosis, which is mediated by the hepatic lipogenic pathway, primarily through SREBP-1c, causing hepatic steatosis and hyperlipidemia. SREBP-1c regulates the expression of lipogenic gene FAS [39]. Therefore, inhibition of LXR-α expression may be beneficial for the control of NAFLD. Administration ofSCPE could significantly inhibit LXR-α, SREBP-1c and FAS gene expression, and offer effective treatment for high fat dietinduced NAFLD. 

Studies in recent years have demonstrated that ROS produced during fatty acid oxidation triggers inflammatory pathway involving NF-kB, TNF-α, IL-6, leading to hepatic inflammation [40,41,42]. Therefore, steatohepatitis is atypical characteristic of NAFLD. Some NAFLD can develop into fibrosis, and even malignant transformation (hepatocellular carcinoma) [43]. In this study, we observed that HFD mice supplemented with SCPE had decreased liver weight, ameliorated hepatic lipid accumulation, and reduced ALT activity compared to the HFD group. In addition, the increase of serum TNF-α, IL-6 was completely inhibited and the expression of liver inflammation genes wassignificantly down-regulated after treatment with SCPE for 8 weeks. The inhibitory effectof SCPE on NAFLD wasfurther confirmed by conventional histological assessment of livers and adipocytes. 

These antioxidant effects, reducing adiposity and preventingNAFLD should be attributedto the abundant phenolic compound in SCPE, which reaches 102 mg GA/g. To identify the phenolic compound in SCPE, UHPLC-Q-Orbitrap-HRMS was used. Twelve phenolic compounds were identified, in which rutin, kaempferol 3-O-beta-rutinoside and quercetin 3-beta-D-glucoside were flavonoid derivatives. The compounds of No. 8–12 were flavonoids and the others were phenolic acid. Naringenin, rutin and chrysin were the main phenolic compounds in SCPE. All of those phenolic compounds lead to SCPE possessinghigher antioxidant capability. The FRAP of SCPE was 297.80 mg Trolox/g, obviously higher than that of the 13 bee pollens from Turkey, which rangedfrom 11.77 to 105.06 μmolTrolox/g pollen (equating to 2.94–26.42 mg Trolox/g pollen) [44]. The IC_50_ of DPPH scavenging activity of SCPE was 0.74 mg/mL, lower than that of water extract (2.36 mg/mL) or methanol extract (1.72 mg/mL) of linder bee pollen [45]. Accordingly, SCPE is rich in phenolic compounds, which can attenuate oxidative stress and inhibit inflammationby scavenging ROS in obese mice. Furthermore, adiponectin, whose expression and secretion are unique to differentiated adipocytes, can regulate energy homeostasis, glucose and lipid metabolism and exert anti-inflammatory action [32]. Liver adiponectin secretion was inhibited by HFD and increased after SCPE consumption, which couldbe another means by which SCPE can reduce adiposity and prevent NAFLD. Briefly, SCPE rich in phenolic compounds reduces adiposity and prevents NAFLD by scavenging ROS, attenuating oxidative stress, increasing adiponectin secretion and suppressing inflammation in obese mice.

Gut microbiota, a complex ecosystem of trillions of microorganisms, codevelops with the host from birth and depends on the host genome, nutrition and life-style [46].A balanced gut microbiota composition confers benefits to the host, whereas the disturbance of this microbial composition is associated with chronic metabolic disorder, which is involved in esoenteritis, obesity, diabetes and so on [47].Recent studies have put forward a new concept, “three Ps for health”, which declares probiotics, prebiotics and polyphenols to modulate either the composition or metabolic/immunological activity of the host gut microbiota [48]. It promotes polyphenols to the same biological level asprebiotics [49]. In this study, we observed the prebiotic effect of SCPE on gut microbiota in obese mice. The results showed that 15.72 g/kg BW of SCPE significantly changed the gut microbial composition in obese mice, manifesting as arapid increase of *bifidobacterium* and *biofidobacteriaceae* in SCPE treatment mice (Figure 6II). The abundance of *lactobacillaceae* and *lactobacillus* in obese mice gut ecosystem were remarkably reduced compared to normal mice (Figure 6III,IV), whereas different doses of SCPE consumption increased their abundance with a dose-effect relationship. Both *lactobacillus* and *biofidobacterium* are probiotic bacteria, whose growth and abundance in the gut ecosystem directly influences host health. Phenolic compounds can act as promoting factors of growth, proliferation, or survival for probiotic bacteria, including *lactobacillus* and *biofidobacterium* strains, and therefore, exertprebiotic effects. At the same time, phenolic compounds can inhibit the proliferation of some pathogenic bacteria [50]. As expected, the growth of *pseudomonas* was inhibited by SCPE treatment in this study. 

While polyphenols modulate the gut microbial balance by stimulation of the growth of beneficial bacteria and inhibition of pathogen bacteria, gut microbiota plays an important role in the transformation and assimilation of polyphenols [51]. Gut microbiota can hydrolyze glycosides, glucuronides, esters, lactones and carry out ring-cleavage, decarboxylation, demethylation, and dehydroxylation reaction to metabolize the phenolic compounds to easy assimilated bioactive compound that result in affecting the intestinal ecology and influence host health [52,53]. In this study, rutin (quercetin 3-rhamnosyl (1-6) glucoside) is one of the flavonoid compounds in SCPE, which is only absorbed with difficulty. Rutin could be metabolized and rendered into quercetin when it arrived in the colon and encounters*Lactobacillus* and*Bifidobacterium* strains [54]. Then,quercetin could be degraded by gut microbiota to produce simpler and more easilyassimilated compounds after the C-ring breakdown. Once the C-ring breakdown takes place at different positions, different simpler phenolic compounds can be formed, probably including 2-(3-hydroxyphenyl) acetic acid, 2-(3,4-dihydroxyphenyl) acetic acid, 3,4-dihydroxybenzoic acid and phloroglucinol [54]. In this study, we investigated whether SCPE promotes the growth of some probiotics and inhibitsthe growth of pathogenic bacteria. However, the modulation of the gut microbiota on the degradation of polyphenols in SCPE wasnot explored, and thiswill be a focus in the following study. 

## 5. Conclusions 

In conclusion, SCPE possessing abundant polyphenols and higher antioxidant activities could attenuate body and liver weight gain, decrease fasting blood glucose, cut down the lipid accumulation in serum and liver, lessen oxidative injury and inflammation in obesity mice. Moreover, SCPEcould completelyinhibit the formation of NAFLD by inhibition of LXR-α, SREBP-1c and FAS gene expression, and repair high fat diet induced liver damage. Additionally, SCPE modulates the structural alteration of gut microbiota in obesity mice, which couldbe associated with the polyphenols in SCPE. In particular, SCPE could enhance the relative abundance of Lactobacillus and reduce the relative abundance of the pathogenic bacteria. To further elucidate the mechanisms of modulation of SCPE on gut microbiota, studies on the change of microbial membrane and phenolic-derived metabolites should be carried out. 

## Figures and Tables

**Figure 1 nutrients-11-00346-f001:**
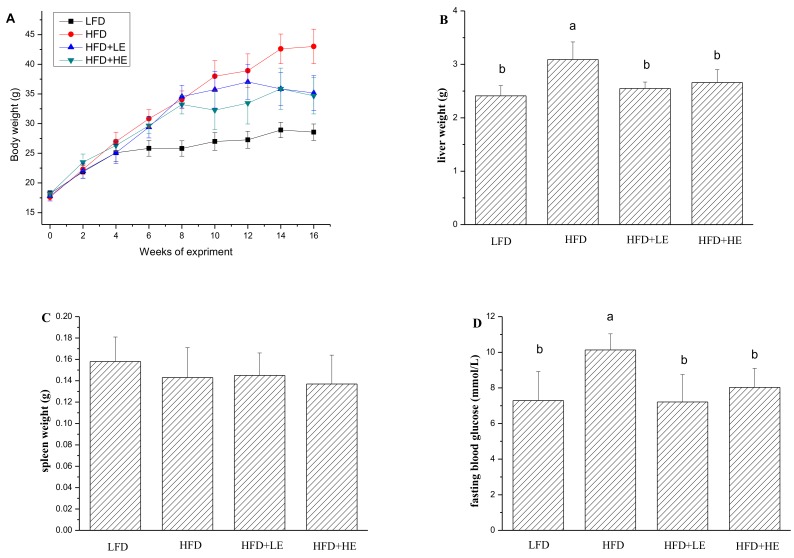
Effects of SCPE on fasting body weight (**A**), liver weight (**B**), spleen weight (**C**) and fasting blood glucose (**D**) of mice. Different lower-case letters correspond to significant differences at *p*< 0.05.

**Figure 2 nutrients-11-00346-f002:**
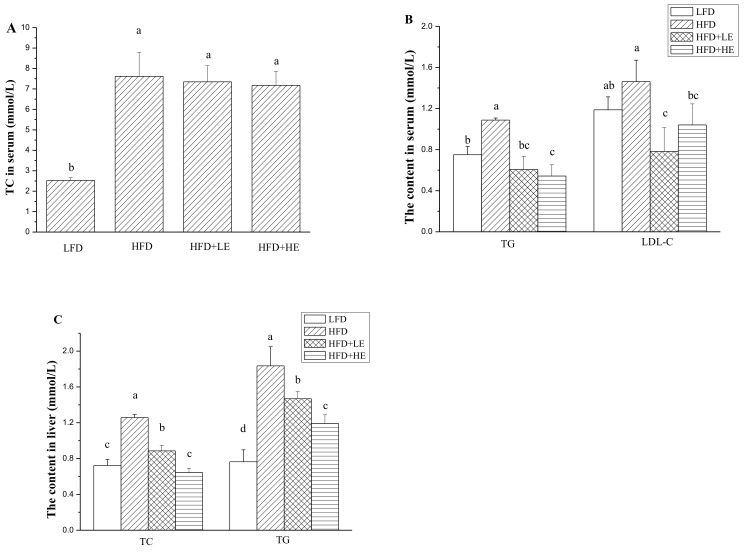
Effects of SCPE on the contents of TC in serum (**A**), TG and LDL-C in serum (**B**), TC and TG in liver (**C**). Different lower-case letters correspond to significant differences at *p*< 0.05.

**Figure 3 nutrients-11-00346-f003:**
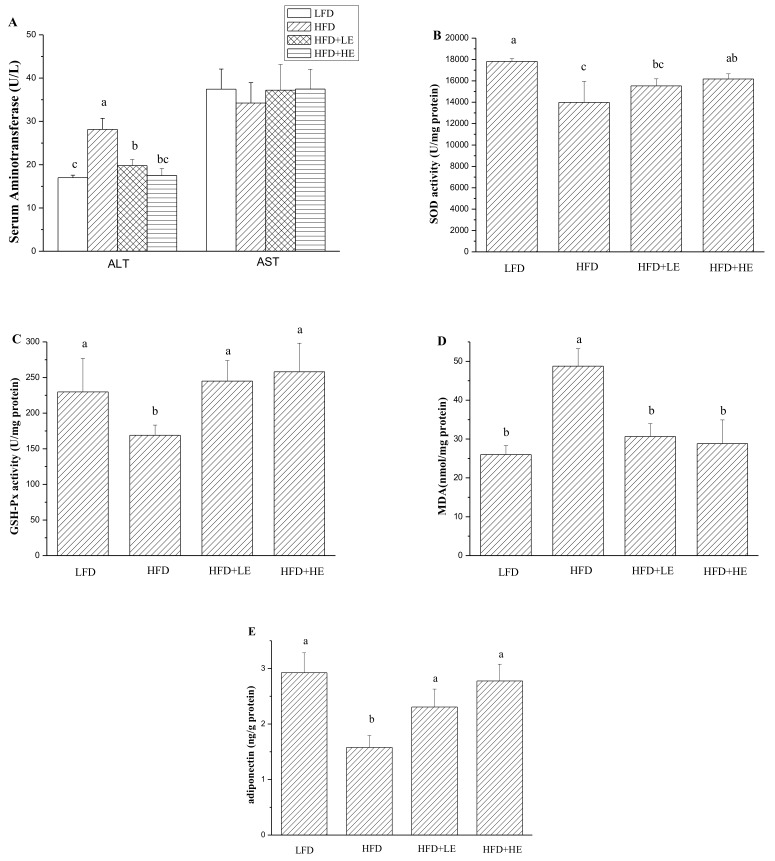
Effects of SCPE on serum ALT and AST activities (**A**), hepatic SOD activity (**B**), GSH-Px (**C**), MDA (**D**) and adiponectin (**E**) content. Different lower-case letters correspond to significant differences at *p*< 0.05.

**Figure 4 nutrients-11-00346-f004:**
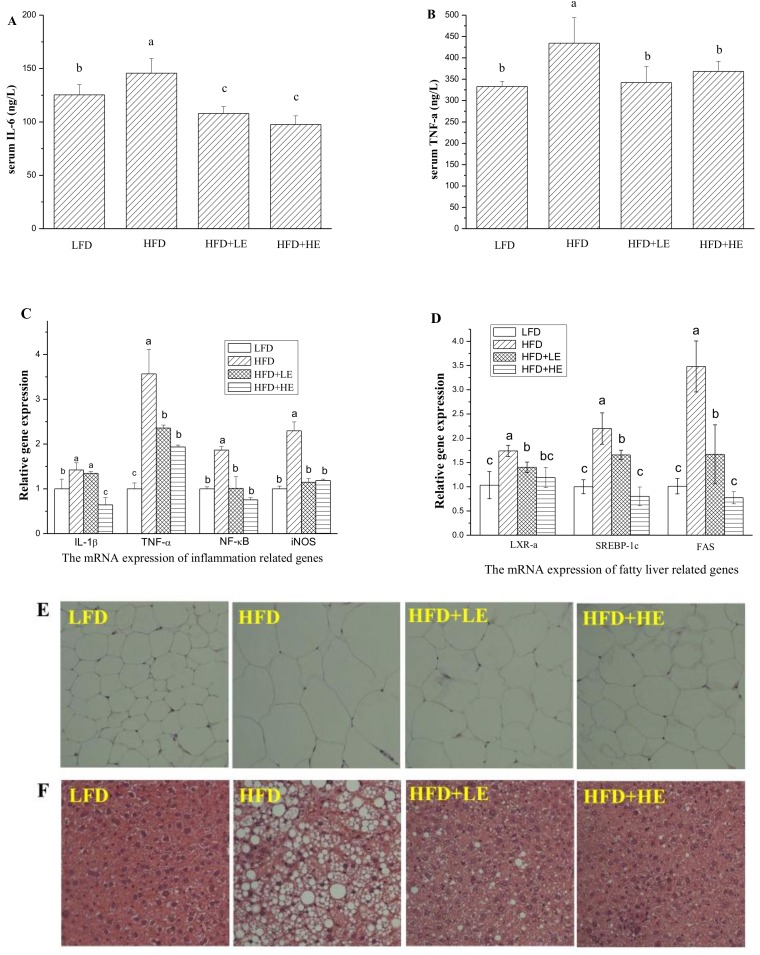
Effects of SCPE on the levels of serum IL-6 (**A**), TNF-α (**B**), inflammation-related genes in liver (**C**), fatty liver-related genes (**D**), and H&E staining of adipose (**E**) (200×) and liver tissue (**F**) (200× magnification). Different lower-case letters correspond to significant differences at *p*< 0.05.

**Figure 5 nutrients-11-00346-f005:**
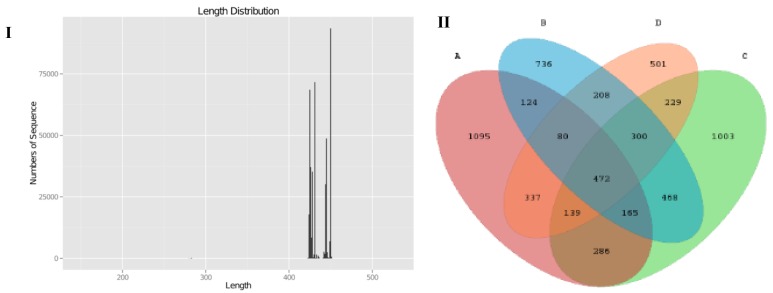
DNA raw data and comparison of the OTUs in all examined colon contents. I: DNA sequencing number and length; II: Venn diagram for describing the common and unique OTUs. A: LFD; B: HFD; C: HFD + LE; D: HFD + HE. Different lower-case letters correspond to significant differences at *p*< 0.05.

**Figure 6 nutrients-11-00346-f006:**
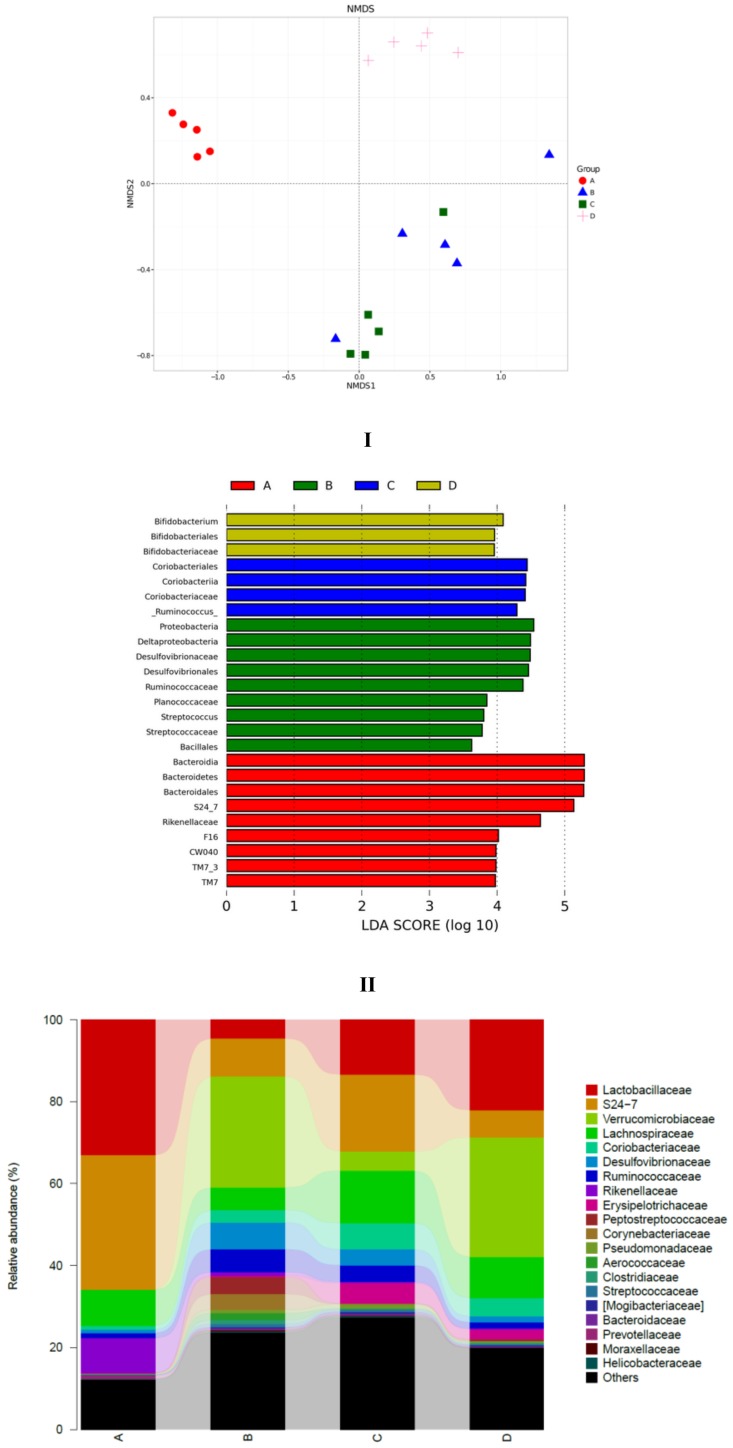
SCPE modulated the HFD-disrupted gut microbiota composition. I: unweightedUniFrac NMD analysis. II: the significantly different species in four groups. III: the relative abundance (%) of bacterial at family level in four groups. IV: Highly differentiated OTU at phylum and genus levels in four groups. A: LFD; B: HFD; C: HFD + LE; D: HFD + HE. Different lower-case letters correspond to significant differences at *p* < 0.05.

**Table 1 nutrients-11-00346-t001:** Characterization of the 12 compounds of SCPE by UHPLC-Q-Orbitrap-HRMS.

No.	Compounds	t_R_ (min)	Exptl (m/z)	Calcd (m/z)	Fragment Ions (m/z)	Content (mg/g)
1	2,3-Dihydroxybenzoic acid	2.26	153.0193	153.0193	135.0 (5), 117.0 (10), 109.0 (10), 91 (5), 73.0 (5)	0.0111
2	4-Hydroxybenzoic acid	2.72	137.0244	137.0244	119.0 (50), 93.0 (10), 75.0 (5)	-
3	2,4-Dihydroxybenzoic Acid	3.20	153.0193	153.0193	135.0 (50), 117.0 (10), 109.0 (50), 91 (5), 73.0 (5)	0.0673
4	5-Methoxysalicylic acid	3.28	167.0349	167.035	149.0 (20), 123.0 (10)	0.0176
5	Rutin	3.49	609.1462	609.1461	301.0 (100)	0.5375
6	Kaempferol 3-O-β-rutinoside	3.62	593.1512	593.1512	285.0 (10)	-
7	Quercetin 3-β-D-glucoside	3.65	463.0883	463.0882	301.1 (10), 179.1(10)	-
8	4-Hydroxycoumarin	3.90	161.0245	161.0244	143.0 (40), 133.0 (10), 117.0 (20), 115.0 (5), 99.0 (5)	-
9	Naringenin	4.84	271.0612	271.0612	271.1 (40), 165.0 (40)	1.8934
10	Apigenin	4.88	269.0455	269.0455	151.0 (15)	-
11	Chrysin	5.73	253.0506	253.0506	151.1 (25)	0.5627
12	Isoliquiritigenin	5.77	255.0663	255.0663	119.0 (20)	0.2830

**Table 2 nutrients-11-00346-t002:** The result of TPC, TFC and antioxidantactivities of SCPE in vitro.

TPC (mg GA/g)	TFC (mg rutin/g)	DPPH Scavenging Activity IC_50_ (mg /mL)	FRAP (mgTrolox/g)	Ferrous Ion-Chelating Activity (mg Na_2_EDTA/g)
101.83 ± 0.01	73.22 ± 0.04	0.74 ± 0.05	297.80 ± 0.92	44.82 ± 0.60

Results presented in the table are expressed as mean ± standard deviation (SD) for 3 replications.

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
