# Peer review of "Impact of SchisandraChinensis Bee Pollen on Nonalcoholic Fatty Liver Disease and Gut Microbiota in HighFat Diet Induced Obese Mice"

_nutrients, 2019, doi:10.3390/nu11020346_

Round 1
Reviewer 1 Report
The authors propose that Phenolic compounds of Schisandra chinensis bee pollen prevent nonalcoholic fatty liver disease and modulates the gut microbiota in high-fat-diet-induced obese mice. The article is well written in certain aspects, scientifically sound but it lacks novelty or originality. However, their attempts to test this do not appear to be adequate or appropriate. The main drawback of this study – Originality or novelty. There are many research article published showing the beneficial effect of Schisandra chinensis in NALFD. The Author has claimed in the title Phenolic compounds of Schisandra chinensis prevents NAFLD, but the author didn’t specify the phenolic compound of Schisandra chinensis.
Major comments
· The main drawback of this study is a novelty, as there many research articles explaining the role of Schisandra chinensis. The author has to explain the differences between this study and other studies.
http://dx.doi.org/10.1016/j.biopha.2017.09.083
http://dx.doi.org/10.1155/2016/6171658
http://dx.doi.org/10.1016/j.jep.2016.03.021
Molecules 2018, 23, 352; doi:10.3390/molecules23020352
http://dx.doi.org/10.1016/j.nutres.2017.07.002
Wang et al. Lipids in Health and Disease (2016) 15:195
· In the title, the author has claimed phenolic compounds of Schisandra chinensis prevents NAFLD, but in the article, the author has explained about the SCPE. Either author has to specify phenolic compounds or remove the word phenolic compounds from the title.
· If the author claims SCPE extract contains only the phenolic compounds, then the author should add additional group where mice fed HFD treated with normal Schisandra chinensis extract.
· Statistical analysis is poorly explained. The author has to explain what kind of statistical test is performed for each experiment – Student T-test or one-way ANOVA or two-way ANOVA.
· Figure legends explained inadequately; statistics should be defined in each figure legends and use Greek symbols to describe the statistics between the groups. The author has used a or b in the figure panels but didn’t explain what does a or b stands for in the figure legend. The author should explain their results more scientifically and precisely
· In Figure 2, panel B TG levels and LDL-C levels are lower in HFD-LE and HFD-HE compared to LFD. What will be the effect of SCPE extract on mice fed with LFD? The author needs to provide the results, additional groups, where LFD fed mice were treated with SCPE extract compare the results with C57BL/6 mice fed in normal chow data.
· In Figure 2 for statistics comparison, the author has mentioned b, a, bc, c bc but the author failed to explain statistics comparison in the figure legends.
· In general, for NAFLD mice TG levels will be more the 3 mmol/L and LDL-C levels will be more than 2 mmol/L. Can the author explain why they observed lower TG and LDL-C levels in mice fed with HFD?
· In figure 3, the Author talks about liver Damage. ALT and AST levels are considered as “gold standard” markers for hepatic cell injury. Only the ALT level is elevated not the AST levels. Just based on one parameter we can’t come to any conclusion. The Author should provide AST/ALT ratio, evaluate liver damage score.
· Hepatic lipogenesis genes such as LXR-alpha, SREBP-1, FAS, and ACC will be increased during the NAFLD or in obese mice. It will be more interesting if the author provides the data regarding hepatic lipogenesis and effect of SCPE extract on those genes or proteins.
· Acetylation plays a vital role in the development of obesity-mediated NAFLD. Proteomic studies have shown the presence of hyperacetylated proteins in the livers of high-fat diet fed mice. It would be interesting if the author shows protein acetylation and effect of SCPE on protein acetylation.
· Liver fibrosis is another hallmark marker for liver damage and NAFLD. The author should provide data regarding the liver fibrosis – histological level and at molecular levels.
· The Author didn’t talk about the apoptosis in the liver. The author should explain what the effect of SCPE extracts on apoptosis in the liver cells.
Author Response
Dear Reviewer:
We are truly grateful to your critical comments and thoughtful suggestions. We have made careful modifications in the original manuscript. All changes were in red color. We wish that the new manuscript could meet your standard. Our point-by-point responses were shown as follows.
Major comments
Q1: The main drawback of this study is a novelty, as there many research articles explaining the role of Schisandra chinensis. The author has to explain the differences between this study and other studies.
http://dx.doi.org/10.1016/j.biopha.2017.09.083
http://dx.doi.org/10.1155/2016/6171658
http://dx.doi.org/10.1016/j.jep.2016.03.021
Molecules 2018, 23, 352; doi:10.3390/molecules23020352
http://dx.doi.org/10.1016/j.nutres.2017.07.002
Wang et al. Lipids in Health and Disease (2016) 15:195
A1: The first research article, “Protective effect of Schisandra chinensis bee pollen extract on liver and kidney injury induced by cisplatin in rats”, discussed the protective effect of Schisandra chinensis bee pollen extract from liver damage-induced by cisplatin. Our manuscript focused on prevention of Schisandra chinensis bee pollen extract from NAFLD induced by high fat diet. We identified the phenolic compounds of Schisandra chinensis bee pollen to clarify the bioactive constituents. Moreover, we found that gut microbiota plays important role in reducing body weight and preventing from NAFLD during Schisandra chinensis bee pollen consumption.
The second research article, “Schisandrin B: A Double-Edged Sword in Nonalcoholic Fatty Liver Disease” discussed the effect of an active dibenzoo ctadiene lignan, Schisandrin B, on NAFLD. Schisandrin B, isolated from the fruit of Schisandra chinensis, has not been found in the Schisandra chinensis bee pollen until now. We analysis the extract of Schisandra chinensis bee pollen. The objects of the two studies were different.
The third article is “Schisandra chinensis extract ameliorates nonalcoholic fatty liver via inhibition of endoplasmic reticulum stress”. The object of this study is the Schisandra chinensis extract. And the mechanism of that Schisandra chinensis extract ameliorates nonalcoholic fatty liver is inhibiting endoplasmic reticulum stress. However, the object of our study is Schisandra chinensis bee pollen.
The first fourth article, “Hepatoprotective Effects of a Functional Formula of Three Chinese Medicinal Herbs: Experimental Evidence and Network Pharmacology-Based Identification of Mechanism of Action and Potential Bioactive Components” evaluated the effect of Jian-Gan-Bao, a functional herbal formula, consists of Schisandra chinensis on acute liver damage and NAFLD. However, the object of our study is Schisandra chinensis bee pollen.
The fifth article is “Schisandra chinensis berry extract protects against steatosis by inhibiting histone acetylation in oleic acid–treated HepG2 cells and in the livers of diet-induced obese mice”. This article showed that Schisandra chinensis berry can be used as a novel therapeutic agent for prevention of steatosis. However, the object of our study is Schisandra chinensis bee pollen.
The sixth article, “Schisandra polysaccharide inhibits hepatic lipid accumulation by downregulating expression of SREBPs in NAFLD mice”, studied the effect of Schisandra polysaccharide on NAFLD. However, polysaccharide was not involved in our study.
In short, only the first of the above-mentioned studies discussed the Schisandra chinensis bee pollen. But it did not refer to NAFLD and gut microbiota. The objects other studies are not Schisandra chinensis bee pollen. Schisandra chinensis bee pollen has been used as a healthy food in China for centuries, but the bioactive compounds of Schisandra chinensis bee pollen have not been studied clearly up to now. And the mechanism of prevention some diseases is unknown. Therefore, we identified the phenolic compounds in Schisandra chinensis bee pollen, and evaluated its effect on reducing body weight and NAFLD. Moreover, gut microbiota was analyzed to explain the good effect of Schisandra chinensis bee pollen.
Q2: In the title, the author has claimed phenolic compounds of Schisandra chinensis prevents NAFLD, but in the article, the author has explained about the SCPE. Either author has to specify phenolic compounds or remove the word phenolic compounds from the title.
A2: The title has been revised as “Impact of Schisandra chinensis bee pollen on nonalcoholic fatty liver disease and gut microbiota in high fat diet induced obese mice”.
Q3: If the author claims SCPE extract contains only the phenolic compounds, then the author should add additional group where mice fed HFD treated with normal Schisandra chinensis extract.
A3: SCPE contains phenolic compounds, sugars, proteins, pigments, mineral elements and so on. In our study, the antioxidant activities of SCPE should be ascribed to phenolic compounds. Therefore, phenolic compounds were emphasized in the manuscript.
Q4: Statistical analysis is poorly explained. The author has to explain what kind of statistical test is performed for each experiment – Student T-test or one-way ANOVA or two-way ANOVA.
A4: One-way ANOVA were performed for the statistical test of each experiment. Related description was added at 2.10 of the manuscript.
Q5: Figure legends explained inadequately; statistics should be defined in each figure legends and use Greek symbols to describe the statistics between the groups. The author has used a or b in the figure panels but didn’t explain what does a or b stands for in the figure legend. The author should explain their results more scientifically and precisely.
A5: “Different lower-case letters correspond to significant differences at p < 0.05.” has been added at the end of all figure legends.
Q6: In Figure 2, panel B TG levels and LDL-C levels are lower in HFD-LE and HFD-HE compared to LFD. What will be the effect of SCPE extract on mice fed with LFD? The author needs to provide the results, additional groups, where LFD fed mice were treated with SCPE extract compare the results with C57BL/6 mice fed in normal chow data.
A6: We think that if the mice fed with LFD+SCPE, TG levels and LDL-C levels will be reduced compared with LFD only. Certainly, that LFD+SCPE (positive control) were added in our study will make it more perfect and more persuasive. However, there are no positive control in some other studies. Several relevant literatures are as follows:
Wu, T.; Yin, J.; Zhang, G.; et al., Mulberry and cherry anthocyanin consumption prevents oxidative stress and inflammation in diet-induced obese mice. Mol. Nutr. Food Res. 2015, 00, 1-8.
Jurado-Ruiz E.; Varela L. M.; Luque A.; et al., An extra virgin olive oil rich diet intervention ameliorates the nonalcoholic steatohepatitis induced by a high-fat “western-type” diet in mice. Mol. Nutr. Food Res. 2017, 61(3) 1600549
Moura M. H. C.; Cunha M. G.; Alezandro M. R.; et al., Phenolic-rich jaboticaba (Plinia jaboticaba (Vell.) Berg) extracts prevent high-fat-sucrose diet-induced obesity in C57BL/6 mice. Food Res. Int. 2018, 107, 48-60.
Additionally, we need 8 weeks to supplement group (LFD+SCPE). Only 10 days were permitted for revised the manuscript. We are sorry to cannot add LFD+SCPE groups in this study. And we think positive control will be designed in our future studies.
Q7: In Figure 2 for statistics comparison, the author has mentioned b, a, bc, c bc but the author failed to explain statistics comparison in the figure legends.
A7: “Different lower-case letters correspond to significant differences at p < 0.05.” has been added in figure 2 legend.
Q8: In general, for NAFLD mice TG levels will be more the 3 mmol/L and LDL-C levels will be more than 2 mmol/L. Can the author explain why they observed lower TG and LDL-C levels in mice fed with HFD?
A8: In other studies, serum TG for NAFLD mice were also determined and the values were different.
“Triglyceride is strongly associated with nonalcoholic fatty liver disease among markers of hyperlipidemia and diabetes” DOI: 10.3892/br.2014.309. The value of serum TG for NAFLD mice was 104 mg/dl, equating with 1.17 mmol/l.
“Mulberry and cherry anthocyanin consumption prevents oxidative stress and inflammation in diet-induced obese mice” DOI 10.1002/mnfr.201500734. The value of serum TG for mice feeding with high fat diet for 16 weeks was 1.5 mmol/ml.
In different studies, the method for determining serum TG was different. Maybe, systematical error existed in different methods. The species of mice may be another reason for the difference. Therefore, a little lower value for TG and LDL-C levels occurred in our studies.
Q9: In figure 3, the Author talks about liver Damage. ALT and AST levels are considered as “gold standard” markers for hepatic cell injury. Only the ALT level is elevated not the AST levels. Just based on one parameter we can’t come to any conclusion. The Author should provide AST/ALT ratio, evaluate liver damage score.
A9: Usually, ALT is sensitive to the progress of NAFLD, and the change of AST is mild or no greater than ALT. Therefore, an significant increase was found in ALT of HFD group, and AST of all groups did not differ significantly. We read several relevant literatures and understand that AST/ALT ratio is a good index for evaluate liver damage, indeed. However, some studies on NAFLD did not afford the AST/ALT ratio, which are as follows:
Melatonin improves non-alcoholic fatty liver disease via MAPK-JNK/P38 signaling in high-fat-diet-induced obese mice. DOI 10.1186/s12944-016-0370-9
Schisandra polysaccharide inhibits hepatic lipid accumulation by downregulating expression of SREBPs in NAFLD mice. DOI 10.1186/s12944-016-0358-5
Gallic Acid Ameliorated Impaired Glucose and Lipid Homeostasis in High Fat Diet-Induced NAFLD Mice. doi:10.1371/journal.pone.0096969
Therefore, we think that AST/ALT ratio did not afford is also OK.
Q10: Hepatic lipogenesis genes such as LXR-alpha, SREBP-1, FAS, and ACC will be increased during the NAFLD or in obese mice. It will be more interesting if the author provides the data regarding hepatic lipogenesis and effect of SCPE extract on those genes or proteins.
A10: Thank you for the good suggestion. LXR-alpha, SREBP-1, FAS, and ACC are good genes for evaluating the hepatic lipogenesis. However, we are sorry that we cannot provide these data because the limited time for revision. These relevant genes will be tested in our future studies of NAFLD.
Q11: Acetylation plays a vital role in the development of obesity-mediated NAFLD. Proteomic studies have shown the presence of hyperacetylated proteins in the livers of high-fat diet fed mice. It would be interesting if the author shows protein acetylation and effect of SCPE on protein acetylation.
A11: In this study, we preliminarily explored the effect of SCPE on reducing body weight, preventing NAFLD and regulating gut microbiota. The detailed mechanism including protein acetylation will be explored in the next study.
Q12: Liver fibrosis is another hallmark marker for liver damage and NAFLD. The author should provide data regarding the liver fibrosis – histological level and at molecular levels.
A12: The emphases of this study are the effect of SCPE on prevention from NAFLD and regulation of gut microbiota. The mechanism of SCPE prevent from NAFLD will be the next study. And liver fibrosis at histological level and molecular levels will be explored in the next study.
Q13: The Author didn’t talk about the apoptosis in the liver. The author should explain what the effect of SCPE extracts on apoptosis in the liver cells.
A13: Administration with SCPE in mice, NAFLD induced by high fat diet was be inhibited and the liver function refreshed, manifesting as the reducing ALT activity and liver weight, normal liver morphology. The apoptosis in the liver was not the object of this study.
We would like to extend our heartfelt thanks for your valuable suggestions to our paper. We have revised the paper in accordance with your suggestions. We would like to also have our revised paper at your disposal, and we would expect to have our paper to be published in “Nutrients” as early as possible. Again, many thanks for your great efforts make to review our paper.
Yours sincerely,Ni Cheng

Reviewer 2 Report
In this paper the authors aim to demonstrate that bee pollen extracts ameliorate metabolic disorders induce by High Fat Feeding. The findings appear interesting and could be of interest. I have however several concerns before final acceptance of the paper.
First, the paper is very dense and the writing very confusing. It is difficult to read. Especially, the authors spend the time at the beginning of the paper to define the abbreviations used for each mice treatment such as LFD, HFD, HFD-HE or LE. However, in the paper they use the terms of control for LFD and model for HFD. Please clarify.
In addition, there is not figure legend and the explanation of the figures are totally deficient. The letter on the bars seem to refer to statistical analysis but I could not find any indication on their significance. In addition, some graphs such as figure 1A does not indicate any statistical analysis.
Finally, it is not indicated how many mice per group are used. This is central for the study in order to validate the significance of the results.
Main scientific concerns:
There is absolutely no justification for the doses of SCPE used as well as no indication for the time of treatment. The authors need to justify this.
Despite the fact that the authors showed their HFD treatment induced a clear hepatic steatosis and a clear increased in the size of the adipocytes, the liver weight is barely increased, and the AST marker is not changed. The changes in classical metabolic markers that we are expected to see increased upon 8 weeks HFD feeding are in general not strongly modify that make me feel that the HFD model is not clearly established. I suggest to the authors to clarify these points.
The authors showed that there is no change in total cholesterol in blood but a decrease in LDL upon SCPE treatment. This is interesting, but the authors need to also measure HDL to confirm the positive effect of their extracts and to explained why there is no changes in TC.
Minor concerns:
The authors cannot say that NAFLD is companied with obesity and Type II diabetes. It is the opposite!
Several abbreviations used across the text need to be defined.
Author Response
Dear Reviewer:
We are truly grateful to your critical comments and thoughtful suggestions. We have made careful modifications in the original manuscript. All changes were in red color. We wish that the new manuscript could meet your standard. Our point-by-point responses were shown as follows.
Q1: First, the paper is very dense and the writing very confusing. It is difficult to read. Especially, the authors spend the time at the beginning of the paper to define the abbreviations used for each mice treatment such as LFD, HFD, HFD-HE or LE. However, in the paper they use the terms of control for LFD and model for HFD. Please clarify.
A1: We have asked an English teacher to revised language of the manuscript. All changes are in red color. The four groups in this study were LFD group, served as control, feeding with low fat diet. HFD group feeding with high fat diet. HFD+LE group feeding with high fat diet plus intragastric administration with 7.86 g SCPE/kg BW. HFD+HE group feeding with high fat diet plus intragastric administration with 15.72 g SCPE/kg BW. Therefore, LFD, HFD, HFD+LE, HFD+HE groups have been used throughout the manuscript.
Q2: In addition, there is not figure legend and the explanation of the figures are totally deficient. The letter on the bars seem to refer to statistical analysis but I could not find any indication on their significance. In addition, some graphs such as figure 1A does not indicate any statistical analysis.
A2: The total figure legends are in page 29. “Different lower-case letters in the bars correspond to significant differences at p < 0.05.” has been added in all figure legends. In figure 1A, statistical analysis was not evaluated because it will be confused if all data marked with different lower-case letters. From figure 1A, the change of body weight during the experiment can be clearly obtained.
Q3: Finally, it is not indicated how many mice per group are used. This is central for the study in order to validate the significance of the results.
A3: At the last line of page 5, 48 male C57BL/6 mice (16-18 g) were divided into four groups in this study. It means there is 12 mice per group.
Main scientific concerns:
Q4: There is absolutely no justification for the doses of SCPE used as well as no indication for the time of treatment. The authors need to justify this.
A4: In our previous study, the protection of SCPE from CCl4-induced liver damage in mice was conducted (Cheng et al., Food and Chemical Toxicology 55 (2013) 234–240 http://dx.doi.org/10.1016/j.fct.2012.11.022). We found that 10 g SCPE /kg BW can protect liver from damage. Therefore, we design the doses of SCPE at 7.86 and 15.72 g/kg BW in this study. In order to establish the obese and NAFLD model, high fat diet was given to mice for 8 weeks. During the SCPE administration, at the end of the 4th week, 6th week and 8th week, we determined the serum ALT activity and body weight to judge the effect of SCPE on reducing body weight and prevention NAFLD.
Q5: Despite the fact that the authors showed their HFD treatment induced a clear hepatic steatosis and a clear increased in the size of the adipocytes, the liver weight is barely increased, and the AST marker is not changed. The changes in classical metabolic markers that we are expected to see increased upon 8 weeks HFD feeding are in general not strongly modify that make me feel that the HFD model is not clearly established. I suggest to the authors to clarify these points.
A5: Usually, ALT is sensitive to the progress of NAFLD, and the change of AST is mild or no greater than ALT. Therefore, an significant increase was found in ALT of HFD group, and AST of all groups did not differ significantly. That AST marker is not changed was found in other relevant literatures, which are as follows:
Melatonin improves non-alcoholic fatty liver disease via MAPK-JNK/P38 signaling in high-fat-diet-induced obese mice. DOI 10.1186/s12944-016-0370-9
Curcumin regulates endogenous and exogenous metabolism via Nrf2-FXRLXR pathway in NAFLD mice. https://doi.org/10.1016/j.biopha.2018.05.135
Moreover, lipid levels and liver histopathological can be used to ascertain the formation of NAFLD.
Q6: The authors showed that there is no change in total cholesterol in blood but a decrease in LDL upon SCPE treatment. This is interesting, but the authors need to also measure HDL to confirm the positive effect of their extracts and to explained why there is no changes in TC.
A6: TC mainly include LDL and HDL, and few of VLDL etc. If the TC level is unchangeable, LDL increased, and then HDL will be decreased. Therefore, we did not determine the HDL levels in this study. Certainly, monitoring both LDL and HDL levels is better than one of them. However, we are sorry not to be able to measure HDL in serum because the retained sample is overdue and can’t be used to determine serum HDL levels. This is a little fault in our study. We will design experiment more carefully in our future studies.
Minor concerns:
Q7: The authors cannot say that NAFLD is companied with obesity and Type II diabetes. It is the opposite!
A7: The error expression has been revised as “Obesity and type 2 diabetes are the two major risk factors for NAFLD, …”
Q8: Several abbreviations used across the text need to be defined.
A8: We have defined the abbreviations.
We would like to extend our heartfelt thanks for your valuable suggestions to our paper. We have revised the paper in accordance with your suggestions. We would like to also have our revised paper at your disposal, and we would expect to have our paper to be published in “Nutrients” as early as possible. Again, many thanks for your great efforts make to review our paper.
Yours sincerely,Ni Cheng

Round 2
Reviewer 1 Report
Dear Authors,
The Authors have provided answers to each question by citing examples of the paper published in other journals, but there is no scientific explanation for from them.
They repeatedly mentioned scope or objective of this study is an effect of SCPE on prevention from NAFLD and regulation of gut microbiota. But they failed to show how SCPE is preventing the NAFLD. There is no proper scientific explanation regarding the molecular mechanism or apoptosis or fibrosis. If the Author requires more time to perform the experiment, They should inform the editor regarding this and get more time to perform the experiment rather than giving this as the reason for not performing the experiments.
Regarding the Statistics comparison, the author's response was not sufficient. They have mentioned that " Different lower-case letters correspond to significant differences at p<0.05" they didn't explain what is a, b, bc? In figure 1, panel B. a is mentioned on top of LFD and HFD group. Does a refer the comparison between LFD and HFD or with other groups? They need to mention for example as "a = LFD vs. HFD"; "b = HFD Vs. HFD+LE"
Author Response
Dear Reviewer:
We are truly grateful to your critical comments and thoughtful suggestions. We have made careful modifications in the original manuscript. All changes were in red color. We wish that the new manuscript could meet your standard. Our point-by-point responses were shown as follows.
Q1: They repeatedly mentioned scope or objective of this study is an effect of SCPE on prevention from NAFLD and regulation of gut microbiota. But they failed to show how SCPE is preventing the NAFLD. There is no proper scientific explanation regarding the molecular mechanism or apoptosis or fibrosis. If the Author requires more time to perform the experiment, They should inform the editor regarding this and get more time to perform the experiment rather than giving this as the reason for not performing the experiments.
A1: LXR-α, SREBP-1c and FAS genes expression in liver have been quantified by Q-PCR, and the results are shown in Fig. 4D. The mechanism of SCPE prevent NAFLD has been added in “Discussion”. Because the main object of this study is to probe the effect of SCPE on reducing body weight and prevention from NAFLD by modulation the gut microbiota. And the detailed molecular mechanism of prevention from NAFLD was the object in our next study. Therefore, only the main genes related with the formation of NAFLD were monitored in this study and the simple mechanism was discussed.
Fig. 4
Q2: Regarding the Statistics comparison, the author's response was not sufficient. They have mentioned that " Different lower-case letters correspond to significant differences at p<0.05" they didn't explain what is a, b, bc? In figure 1, panel B. a is mentioned on top of LFD and HFD group. Does a refer the comparison between LFD and HFD or with other groups? They need to mention for example as "a = LFD vs. HFD"; "b = HFD Vs. HFD+LE"
A2: Regarding the statistics comparison, "Different lower-case letters correspond to significant differences at p<0.05" means if the same lower-case letter exist between any two groups, the difference is not significant. For example, in figure 3 panel B, LFD is “a” and HFD is “c”, which means there is a significant difference between LFD and HFD. LFD is “a” and HFD+HE is “ab”, which means there is not a significant difference between LFD and HFD+HE. HFD is “c” and HFD+HE is ab, which means there is a significant difference between HFD and HFD+HE, and so on.
In brief, if the same lower-case letter exist between any two groups, the difference is not significant. In other words, if there is not the same lower-case letter exist between any two groups, the difference is significant. This expression can compare the statistical difference between any two groups.
We would like to extend our heartfelt thanks for your valuable suggestions to our paper. We have revised the paper in accordance with your suggestions. We would like to also have our revised paper at your disposal, and we would expect to have our paper to be published in “Nutrients” as early as possible. Again, many thanks for your great efforts make to review our paper.
Yours sincerely,Ni Cheng

Reviewer 2 Report
Most of my previous suggestions were taken into consideration
Author Response
Dear reviewer,
We are truly grateful to your hard work on our manuscript. Thank you very much.
Yours sincerely,Ni Cheng